# The Development and Demonstration of a Semi-Automated Regional Hazard Mapping Tool for Tailings Storage Facility Failures

**Sally Innis** [1,*] ⬛, **Negar Ghahramani** [2], **Nahyan Rana** [3], **Scott McDougall** [2], **Stephen G. Evans** [3], **W. Andy Take** [4] and **Nadja C. Kunz** [1,5]

1   Norman B. Keevil Institute of Mining Engineering, The University of British Columbia, Vancouver, BC V6T 1Z4, Canada
2   Department of Earth, Ocean and Atmospheric Sciences, The University of British Columbia, Vancouver, BC V6T 1Z4, Canada
3   Department of Earth and Environmental Sciences, University of Waterloo, Waterloo, ON W2L 3G1, Canada
4   Department of Civil Engineering, Queen's University, Kingston, ON K7L 3N6, Canada
5   School of Public Policy & Global Affairs, The University of British Columbia, Vancouver, BC V6T 1Z4, Canada
*   Correspondence: sally.innis@ubc.ca

**Abstract:** Tailings flows resulting from tailings storage facility (TSF) failures can pose major risks to downstream populations, infrastructure and ecosystems, as evidenced by the 2019 Feijão disaster in Brazil. The development of predictive relationships between tailings flow volume and inundation area is a crucial step in risk assessment by enabling the delineation of hazard zones downstream of a TSF site. This study presents a first-order methodology to investigate downstream areas with the potential of being impacted by tailings flows by recalibrating LAHARZ, a Geographic Information System (GIS)-based computer program originally developed for the inundation area mapping of lahars. The updated model, LAHARZ-T, uses empirical equations to predict inundated valley planimetric and cross-sectional areas as a function of the tailings flow volume. A demonstration of a regional application of the LAHARZ-T model is completed for 46 TSFs across Canada. Although the variability in tailings properties and site characteristics cannot be perfectly incorporated or modelled, the LAHARZ-T model offers an efficient method for high-level, regional scale inundation mapping of several potential TSF failure scenarios.

**Keywords:** tailings flows; dam breach; mine waste hazards; risk assessment; tailings

## 1. Introduction

### 1.1. Background

The extraction and processing of economic minerals and metals from the uneconomic host rock generates enormous volumes of rock, effluent and process water that are discharged, commonly in slurry form, into tailings storage facilities (TSFs). These tailings storage facilities (TSFs) are geotechnically designed to impound mine waste, and occasionally wastewater, in perpetuity. However, there has been an upward trend in high-consequence TSF failures over the last fifty years [1,2], leading to growing concerns from mining stakeholders about the risk of TSFs to society and the environment [3]. The failures of TSFs can produce tailings flows that travel substantial distances downstream and result in long-lasting environmental and socioeconomic impacts [4–7], with a recent example being the 2019 Feijão disaster in Brumadinho, Brazil that led to 272 deaths. The breach released 9.7 million cubic meters (M m$^3$) of tailings that inundated an area of ~3 km$^2$ and travelled 9 km before flowing into the Paraopeba river [4,8]. In 2015, the failure of the Fundão TSF (located in the same mining region as Feijão) released 32 M m$^3$ of tailings, caused 19 deaths and severe, long-lasting environmental consequences [9,10]. The Mt.

Polley TSF failure in 2014 in British Columbia, Canada, released 25 M m$^3$ of tailings that flowed into surrounding waterbodies and impacted the livelihoods of local Indigenous communities [3]. These high-profile incidents prompted the development of new global industry standards for TSF management and increased research attention towards improving the state of predictive hazard-risk assessments [4,11–13].

The current state-of-practice for modelling the downstream impacts from a potential TSF failure is to simulate the flow depths, velocities, and arrival times using numerical models. However, numerical models require site-specific data on tailings properties and terrain characteristics, and computationally intensive, costly software [14]. When TSF information is unavailable and/or the use of complex numerical models is not justified, empirical models are useful for creating first-order, screening-level estimates of potential inundation areas [4,13,15,16].

Recent advancements have been made in developing empirical relationships for tailings flows to predict ranges of the potential inundation area, runout distance and travel path angle from flow volume (e.g., Ghahramani et al. [4], Rana et al. [13]). However, there are no automated or semi-automated hazard mapping tools currently in place for regional scale TSF assessment. High-level, regional scale models have potential applications for a range of mining stakeholders who may seek to better understand, analyze, and/or communicate TSF risks. Such models can serve as first-step assessments to be followed by more detailed modelling where required. Two groups of mining stakeholders with documented interest in high-level TSF risk assessments are: (1) institutional mining investors who seek relatively simple tools that can provide a high-level picture of regional or portfolio-level risk exposure for tailings [17] and water-related issues [18,19]; and (2) regulatory enforcement bodies who lack audit prioritization frameworks capable of providing a high-level understanding of comparative consequences across a national portfolio of TSFs [20–22].

*1.2. Scope and Objectives*

The central goal of this study is to address the gap in high-level, regional or portfolio-scale modelling tools by adapting a statistically calibrated, empirical-based GIS program, LAHARZ, to tailings flows. The output of the modified LAHARZ model is a two-dimensional, spatially mapped, probabilistic inundation area that works with publicly available data from the Global Tailings Database (GTD; GRID-Arendal [22]). The LAHARZ model, calibrated to tailings flows and with refinements to the original code including automation of the model given a set of model inputs, is referred to in this paper as LAHARZ-T. The LAHARZ-T model allows users the ability to model multiple tailings flows with limited data, cost, time and computing power. As a demonstration of LAHARZ-T at the regional scale, inundation areas from large Canadian TSFs (dam height $\geq$ 10 m) within the GTD are modelled in order to note the efficiency and limitations of the program.

While LAHARZ-T is aimed at performing screening level assessments at the site level and high-level regional or portfolio-level assessments of potential inundation areas, the deterministic use of empirical models is discouraged at any level of analysis [5,13,23]. Thus, whether LAHARZ-T is used at the site-specific or the regional scale, the model is intended to represent a probabilistic (or semi-probabilistic) methodology. The LAHARZ-T model is not intended to be unconditionally applied in breach-runout analyses, but rather to aid practitioners in conducting screening-level assessments by offering a streamlined approach to estimating potential inundation areas. These studies may subsequently warrant more detailed risk assessments and numerical modelling.

The objectives of this paper are to: (1) examine the applicability of a semi-physical area-volume scaling relationship for the cross-sectional inundation area of tailings flow cases, (2) integrate the scaling relationship results from this study and Ghahramani et al. [4] into LAHARZ to produce the high-level tailings flow model, LAHARZ-T, (3) demonstrate the applicability, methodology and limitations of LAHARZ-T as a regional scale inundation mapping tool by modelling potential areas from 46 large, Canadian TSFs, and (4) assess the

potential utility of the LAHARZ-T model to institutional mining investors and policymakers. This study is an extension of the preliminary work presented in Innis et al. [24].

## 2. Existing Empirical Tools for Modelling Tailings Flows

### 2.1. Empirical Relationships

Potential runout distances and inundation areas of tailings flows are challenging to predict. The variability of TSF configuration, tailings rheology and downstream geomorphology, as well as limited case study information, are some reasons why empirical models for tailings flows are prone to significant uncertainty. Table 1 presents a summary of published empirical relationships for tailings flows.

**Table 1.** Summary of semi-empirical volume—area or runout relationships of tailings flows from preceding studies (updated and modified from Ghahramani et al. [4]). Notations are as follows: $n$ is the number of observations; $R^2$ is the coefficient of determination of the best-fit line; $V_F$ denotes total outflow volume reported in either M m$^3$ by Rico et al. [5] and Concha Larrauri & Lall [23] and reported in this paper, or in m$^3$ by Ghahramani et al. [4] and Rana et al. [13]; $V_T$ is the total reported impounded pre-failure volume reported in M m$^3$ by Rico et al. [5] and Concha Larrauri & Lall [23] and reported in this paper, Ghahramani et al. [4] and Rana et al. in m$^3$; $h$ is the dam height at the breach location in m; $H_f$ from Concha Larrauri & Lall is the dam factor defined as $H(V_F/V_T)V_F$ in M m$^4$; $D_{max}$ from Concha Larrauri & Lall [23] and Rico et al. is the maximum runout distance of the flow in km, whereas $D_{max}$ from Ghahramani et al. [4] and Rana et al. [13] is the Zone 1 runout distance in m; likewise, $A$ from Ghahramani et al. [4] and Rana et al. [13] represents the Zone 1 planimetric inundation area in m$^2$. Piciullo et al. [7] used $R$ to represent total release volume reported in M m$^3$ and $V$ to represent stored volume in M m$^3$. The asterisk (*) indicates equations built into or used in this research.

| Output Parameter | Units | Empirical Relationship | $R^2$ | $n$ | Source |
|---|---|---|---|---|---|
| Total release volume | M m$^3$ | $V_F = 0.354\ V_T{}^{1.01}$ | 0.86 | 22 | Rico et al. [5] |
| Maximum runout distance | m | $D_{max} = 14.45\ V_F{}^{0.76}$ | 0.56 | 23 | Rico et al. [5] |
| Maximum runout distance | m | $D_{max} = 0.05\ h^{1.41}$ | 0.16 | 25 | Rico et al. [5] |
| Maximum runout distance | m | $D_{max} = 1.61\ (h\ V_F)^{0.66}$ | 0.57 | 24 | Rico et al. [5] |
| Total release volume * | M m$^3$ | $V_F = 0.332\ V_T{}^{0.95}$ | 0.89 | 29 | Concha Larrauri & Lall [23] |
| Maximum runout distance | km | $D_{max} = 3.04\ H_F{}^{0.545}$ | 0.53 | 29 | Concha Larrauri & Lall [23] |
| Zone 1 planimetric inundation area * | m$^2$ | $A = 80\ V_F{}^{2/3}$ | 0.57 | 33 | Ghahramani et al. [4] |
| Zone 1 planimetric inundation area: channelized flow | m$^2$ | $A = 14\ V_F{}^{0.81}$ | 0.72 | 22 | Rana et al. [13] |
| Zone 1 runout distance: channelized flow | m | $D = 17\ V_F{}^{0.44}$ | 0.47 | 24 | Rana et al. [13] |
| Zone 1 planimetric inundation area: unconfined flow | m$^2$ | $A = 72\ V_F{}^{0.64}$ | 0.36 | 14 | Rana et al. [13] |
| Zone 1 runout distance: unconfined flow | m | $D = 33\ V_F{}^{0.27}$ | 0.13 | 14 | Rana et al. [13] |
| Total release volume | M m$^3$ | $R = 0.214\ V^{0.35}$ | 0.59 | 70 | Piciullo et al. [7] |

Rico et al. [5] developed a set of empirical relationships that related tailings flow runout distance to other variables such as outflow volume, dam height and a parameter called "dam factor" (dam height multiplied by tailings outflow volume), which was subsequently built on by other authors [7,23,25]. Concha Larrauri and Lall [23] expanded on this work by including newer cases of TSF failures and introduced a new predictor variable, $H_f$, which equals the product of the dam factor and the ratio of tailings outflow volume to tailings storage volume.

Ghahramani et al. [4] introduced a runout zone classification method for tailings flows. The authors defined Zone 1, the primary impact zone, as the extent of the main solid tailings deposit, which is characterized by remotely visible or field-confirmed sedimentation, above typical bankfull elevations if extending into downstream water channels. Zone 2, the

secondary impact zone, was defined as the area downstream of Zone 1 with impacts below typical downstream water levels (i.e., sediment plume impacts) or fluid impacts above typical downstream water levels (i.e., flood wave impacts). The authors investigated the applicability of a semi-physical scaling relationship (Equations (1) and (2)), which was well established for naturally occurring lahars (volcanic debris flows), debris flows and rock avalanches [15,26–29], between total released volume and the Zone 1 planimetric inundation area and maximum cross-sectional inundation area:

$$A = c_A V^{2/3} \tag{1}$$

$$B = c_B V^{2/3} \tag{2}$$

where $A$ is the total planimetric inundation area in $m^2$, $B$ is the maximum cross-sectional inundation area in $m^2$, $V$ is the total flow volume in $m^3$, and $c$ is a dimensionless empirical proportionality coefficient related to flow mobility (i.e., for a given event volume, a higher mobility flow results in a higher planimetric inundation area [15,30]). Equations (1) and (2) are based on the assumption that the areas, $A$ and $B$, likely to be inundated by a lahar, debris flow or rock avalanche only depend on the volume of the flow ($V$) and the topography of the flow path [31]. The calibration of Equations (1) and (2) requires determining best-fit values of the mobility coefficients, $c_A$ and $c_B$, for the flow of interest, in this case tailings flows.

Based on data collected by the geospatial mapping of 33 cases, Ghahramani et al. [4] found that, for a given volume, tailings flows have a planimetric mobility coefficient ($c_A$) of ~80, making them, on average, less mobile than a lahar but more mobile than debris flows and rock avalanches. Rana et al. [13] built on this work by expanding the historical TSF failure database and incorporating site-specific, qualitative uncertainties for channelized and unconfined flows into the empirical space. However, they also cautioned deriving future uses from their empirical relations due to the limited number of case studies with sufficient information on TSF properties and site conditions. These studies also highlighted the importance of probabilistic (rather than deterministic) approaches to tailings flow modelling, as well as the need to perform site-specific investigations to reduce uncertainty [4,13,23].

*2.2. LAHARZ*

LAHARZ is an ArcGIS plug-in program based on the semi-physical scaling relationships shown in Equations (1) and (2). The program was originally developed by the US Geological Survey (USGS) to delineate potential inundation areas of lahars based on one or more user-specified flow volumes [15], and has since been updated to model debris flows and rock avalanches [31]. Bernard et al. [32] recalibrated the mobility coefficients ($c_A$ and $c_B$) for LAHARZ specifically for debris flows in post-wildfire landscapes. To the best of the authors' knowledge, prior to the preliminary research of Innis et al. [24,33], there had not been any previous efforts to apply LAHARZ to model potential tailings flows. Dedring et al. [34] presented a derivative of preliminary research by Innis et al. on the use of LAHARZ for modelling potential tailings flows. In the study by Dedring et al. [34], a cross-sectional mobility coefficient ($c_B$) specific for tailings flows was not investigated and the validation of the model was completed on a single historical case (2019 Feijao, Brazil). The present study builds on Dedring et al. [34] and Innis et al.'s [24,33] previous work by employing a cross-validation method using 32 historical tailings flow cases. Table 2 summarizes derived mobility coefficients available in the literature.

**Table 2.** Comparison of the planimetric and cross-sectional coefficients (dimensionless, $m^2/m^{3(2/3)}$) for different flow types embedded within the LAHARZ program, illustrating the difference in the relative mobility of tailings flows. The asterisk (*) indicates findings derived from this study.

| Flow Type | Planimetric $c$ Coefficient, $c_A$ | Cross-Sectional $c$ Coefficient, $c_B$ | Source |
|---|---|---|---|
| Tailings flows | 80 | * | Ghahramani et al. [4]; * current study. |
| Lahars | 200 | 0.05 | Iverson et al. [15] |
| Debris Flows | 20 | 0.1 | Griswold and Iverson [31] |
| Rock Avalanches | 20 | 0.2 | Griswold and Iverson [31] |

LAHARZ requires a volume estimate of the modelled flow event, an underlying digital elevation model (DEM), and calibrated mobility coefficients to delineate potential inundation areas [30]. The program simulates a downstream flow by using flow direction, flow accumulation and stream delineation grids derived from the input DEM, filling the downstream area cell-by-cell to the calculated cross-sectional area until the calculated planimetric area is reached. The model accounts for deposited material in a mass-balance as it moves downstream. Bulking is assumed negligible in all cases. Flow evolution parameters such as travel time and deposit thickness, which are important output variables in numerical models, are not outputs from LAHARZ. This limitation has implications for the applicability of LAHARZ (and thus LAHARZ-T), as discussed further in Section 5.2.

## 3. Methods

### 3.1. Calibration and Validation of Cross-Sectional Mobility Coefficient ($c_B$)

In this study, the same methodology and failure database presented in Ghahramani et al. [4] was used to investigate the adaptability of the scaling relationship for the cross-sectional inundation area (Equation (2)), as both the planimetric and cross-sectional area relationships are needed to run LAHARZ. A detailed description of planimetric mobility validation is provided in Ghahramani et al. [4].

The Zone 1 maximum cross-sectional area for 32 cases was estimated using the updated tailings dam breach database presented in Ghahramani et al. [4] and Rana et al. [13] (Table 3). The cases are well-documented failures worldwide between 1965 and 2019, covering diversity in geographic location, commodity, failure mechanism, downstream topography and total release volume. The GIS-estimated runout distances for these failures range from 0.1 to 100 km.

The methodology used to calibrate LAHARZ for tailings flows is analogous to the approach used by Iverson et al. [15], Griswold & Iverson [31] and Bernard et al. [32] to calibrate the model for lahars, debris flows, rock avalanches and post-wildfire debris flows. Depending on runout distance, between 3 and 20 cross-sectional lines were defined perpendicular to the Zone 1 travel path. For each case, the cross-sectional area at each reference location was measured using the Shuttle Radar Topography Mission (SRTM) DEM with a resolution of 30 m [35]. The maximum cross-sectional area value was then selected. An example demonstrating the cross-sectional estimation method is shown in Figure 1.

The results of the maximum cross-sectional areas of the 32 cases were used to fit a regression model and examine the adaptability of Equation (2) ($B = c_B V^{2/3}$) for tailings flows. The data were transformed into a logarithmic scale and the standard least-squares linear regression method was then applied. A linear regression model was fit to the data using a specified 2/3 slope.

Ghahramani et al.'s [4] results for the tailings flow specific planimetric mobility coefficient ($c_A = 80$) and the above calculated cross-sectional mobility coefficient were then embedded within the LAHARZ program.

To simulate conditions similar to how the model would run and to determine how well the model would fit potential tailings flows, we adopted a standard leave-one-out cross-validation method [36]. For this test, we used the calibration dataset (Table 3) as the

training dataset and, one at a time, left one TSF failure event out. The planimetric and cross-sectional mobility coefficients of the remaining 31 historical failures were re-calculated and we applied the newly calculated coefficients to the left-out historical failure that was not used to re-calculate the coefficients. This leave-one-out cross-validation method was then repeated for each case in the dataset.

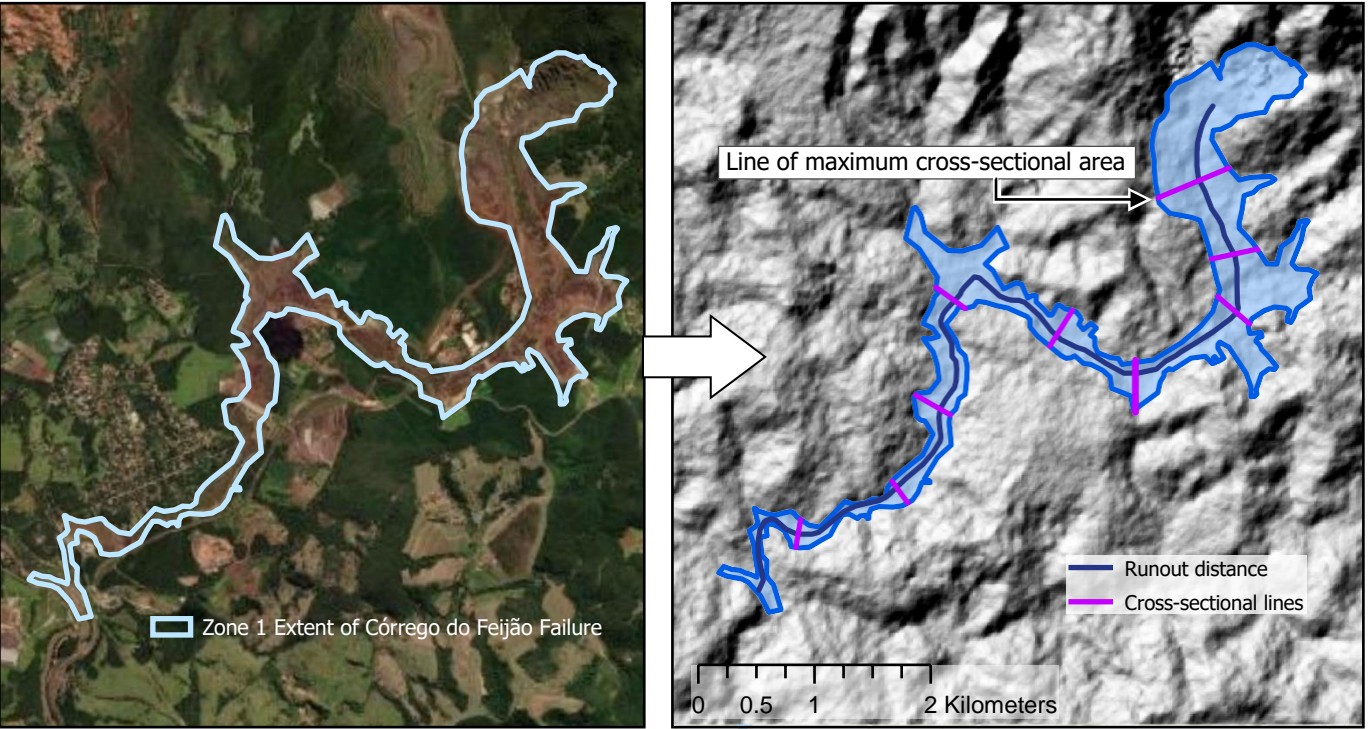

**Figure 1.** Demonstration of the methodology used in this study to estimate Zone 1 maximum cross-sectional area. The blue polygon (**left** and **right**) is the mapped trimline from Rana et al. [13] (including source and inundation area) for the 2019 Feijão tailings flow. The dark blue line (**right**) is the Zone 1 flow path and the purple lines are the cross-sectional lines perpendicular to the flow path. The line of maximum cross-sectional area is noted. The DEM imagery (**right**) is sourced from ©JAXA/METI ALOS PALSAR satellite [37].

*3.2. Development of the Semi-Automated LAHARZ-T Code*

To facilitate application of the LAHARZ-T model for regional scale applicability, the LAHARZ-T program was automated. This allows the user to compile model inputs into a single database and the LAHARZ-T program will automatically run through the database and output the post-failure inundation polygons. This automation allows for multi-site analysis and thereby improves efficiency of the original LAHARZ code, which is a collection of four different manual ArcGIS tools capable of modelling a single site at a time.

**Table 3.** Tailings dam breach case study database (modified from Ghahramani et al. [4] and Rana et al. [13]).

| # | Event | Location | Year | Confined/ Unconfined | Total Released Volume (m$^3$) | Zone 1-Tailings Runout Distance (m) | Zone 1-Inundation Area (m$^2$) | Cross-Sectional Area (max) (m$^2$) |
|---|---|---|---|---|---|---|---|---|
| 1 | Bellavista | Chile | 28 March 1965 | Unconfined | 55,000 | 1300 | 130,000 | 210 |
| 2 | Cerro Negro | Chile | 28 March 1965 | Unconfined | 70,000 | 3200 | 1,300,000 | 710 |
| 3 | El Cobre (New & Old Dams) | Chile | 28 March 1965 | Confined | 2,400,000 | 11,200 | 5,900,000 | 2100 |
| 4 | Los Maquis | Chile | 28 March 1965 | Confined | 21,000 | 1500 | 47,000 | 30 |
| 5 | Sgorigrad | Bulgaria | 1 May 1966 | Confined | 220,000 | 6000 | 400,000 | 940 |
| 6 | Certej | Romania | 30 October 1971 | Confined | 300,000 | 2300 | 380,000 | 810 |
| 7 | Bafokeng | South Africa | 11 November 1974 | Confined | 3,000,000 | 22,000 | 9,000,000 | 1600 |
| 8 | Stava | Italy | 19 July 1985 | Confined | 190,000 | 4200 | 500,000 | 2800 |
| 9 | Stancil | USA | 25 August 1989 | Unconfined | 38,000 | 100 | 7000 | 40 |
| 10 | Tapo Canyon | USA | 17 January 1994 | Confined | 55,000 | 730 | 30,000 | 190 |
| 11 | Merriespruit (Harmony) | South Africa | 22 February 1994 | Unconfined | 600,000 | 2200 | 900,000 | 1200 |
| 12 | Pinto Valley | USA | 22 October 1997 | Confined | 230,000 | 830 | 80,000 | 1200 |
| 13 | Los Frailes/Aznalcollar | Spain | 24 April 1998 | Unconfined | 7,000,000 | 29,000 | 16,000,000 | 2900 |
| 14 | Comurhex, Cogéma/Areva | France | 20 March 2004 | Unconfined | 30,000 | 700 | 70,000 | 220 |
| 15 | Mineracao (Rio Pomba) | Brazil | 10 January 2007 | Confined | 2,000,000 | 40,000 | 8,000,000 | 3800 |
| 16 | Xiangfen | China | 8 September 2008 | Unconfined | 190,000 | 2300 | 400,000 | 690 |
| 17 | Kingston fossil plant | USA | 22 December 2008 | Unconfined | 4,100,000 | 1400 | 800,000 | - |
| 18 | Karamken | Russia | 29 August 2009 | Confined | 2,200,000 | 2900 | 520,000 | 1200 |
| 19 | Las Palmas | Chile | 27 February 2010 | Unconfined | 100,000 | 550 | 80,000 | 340 |
| 20 | Ajka | Hungary | 4 October 2010 | Confined | 1,600,000 | 17,800 | 6,000,000 | 2400 |
| 21 | Kayakari | Japan | 11 March 2011 | Confined | 41,000 | 2000 | 150,000 | 480 |
| 22 | Gullbridge | Canada | 17 December 2012 | Unconfined | 100,500 | 500 | 44,000 | 70 |
| 23 | Obed Mountain | Canada | 31 October 2013 | Confined | 670,000 | 5100 | 1,000,000 | 3600 |
| 24 | Mount Polley | Canada | 4 August 2014 | Confined | 25,600,000 | 9000 | 2,000,000 | 2000 |
| 25 | Fundão | Brazil | 5 November 2015 | Confined | 33,000,000 | 99,000 | 21,000,000 | 7400 |
| 26 | Luoyang | China | 8 August 2016 | Confined | 2,000,000 | 2500 | 300,000 | 1200 |
| 27 | Tonglvshan | China | 12 March 2017 | Unconfined | 500,000 | 500 | 300,000 | 280 |
| 28 | Mishor Rotem | Israel | 30 June 2017 | Confined | 100,000 | 28,000 | 2,000,000 | 1100 |
| 29 | Jharsuguda (Vedanta) | India | 28 August 2017 | Unconfined | 2,600,000 | 640 | 500,000 | 970 |
| 30 | Cieneguita | Mexico | 4 June 2018 | Confined | 440,000 | 15,000 | 500,000 | 920 |
| 31 | Cadia | Australia | 9 March 2018 | Unconfined | 1,330,000 | 480 | 120,000 | 400 |
| 32 | Feijão | Brazil | 25 January 2019 | Confined | 9,650,000 | 9000 | 2,700,000 | 8400 |
| 33 | Cobriza | Peru | 10 July 2019 | Confined | 70,000 | 450 | 70,000 | 120 |

The LAHARZ-T model includes two ArcPy codes. The first code is the semi-automated, tailings specific update to the USGS ArcPy code for LAHARZ. This code, LAHARZ-T, includes the calculated mobility coefficients for tailings flows and runs user-generated inputs from a database, allowing the user to run hundreds of facilities sequentially with efficiency. The code then outputs a raster file of nested hazard maps for each individual failure scenario. While the LAHARZ-T program is automated, the user is still required to provide model inputs for each facility, including a DEM and three text files including the slope of the embankment, one or more user-specified release volumes and one or more user-specified failure initiation points. The resolution requirements for the DEM are discussed in Section 3.3.2. The second code developed for this research is an additional ArcPy code used to batch transform the single raster file nested hazard map output of LAHARZ-T into volume-based polygon files. This code works to improve the efficiency of the model application at the regional scale, as polygon files are more easily geo-processed than raster files.

### 3.3. Demonstration of a LAHARZ-T Application for Regional Scale Analysis

The GTD [22] is presently the most comprehensive disclosure of information on TSFs. The GTD that was commissioned by a group of institutional mining investors following the 2019 Feijão disaster to improve transparency. The database is a compilation of requested information from 106 publicly listed mining companies and, as of August 2022, contains data on the geographic location, height, ownership, status, historical stability, current and projected volumes and other variables for 1862 TSFs at 761 mine sites [22]. It should be noted that the GTD represents a sample of the total number of TSFs worldwide (~20% of active, closed and inactive facilities worldwide [38]).

The following sub-section details the steps to run the LAHARZ-T program at the regional scale by modelling the potential inundation areas from 46 large Canadian TSFs listed in the GTD, with the aim of demonstrating the methodology, potential utility and limitations of the model. Hjorth & Bengtsson [39] classify a large dam as a dam height of $\geq$15 m or $\geq$10 m under certain conditions, including an impounded volume > 15 M m$^3$, a spillway discharge potential of >2000 m$^3$/s or a crest length of <500 m. Since site-specific information on crest length and spillway discharge potential are not freely available from the GTD, all facilities with heights of $\geq$10 m were conservatively included in the large dam classification. It should be noted that this methodology can be applied at different spatial scales, including: (1) across a portfolio of mines, such as an institutional investment portfolio; (2) at the provincial scale to support regulators; or (3) at the global scale for organizations such as the International Council for Mining and Metals (ICMM) to guide risk-informed policy. Given that the Canadian TSFs within the GTD do not include all Canadian facilities, it is important to caution that the results from the demonstration in this paper do not provide a complete hazard profile across all Canadian TSFs.

### 3.3.1. Methodology for LAHARZ-T Regional Scale Analysis

Figure 2 shows a flowchart of the steps in the application of the LAHARZ-T model to regional scale inundation area mapping for a selection of large Canadian TSFs as listed in the GTD [22] and retrieved from Franks et al.'s compilation of the dataset [38].

The proposed methodology in Figure 2 should be viewed as a guideline for future applications; inputs such as the initiation points and release volume should evolve with improved empirical correlations or field knowledge. However, for regional or high-level applications of LAHARZ-T, the following filtering criteria and reasoning are required: (1) exclusion of TSFs with incomplete data on current storage volumes; (2) exclusion of TSFs listed as non-conventional storage types, such as dry-stack, because the LAHARZ-T model is calibrated based on "wet" tailings flows; (3) selection of a single TSF per mine site, based on consequence rating or height, to avoid double counting of downstream risk elements when applied at the regional scale; and (4) exclusion of TSFs with storage volumes of less than 1000 m$^3$.

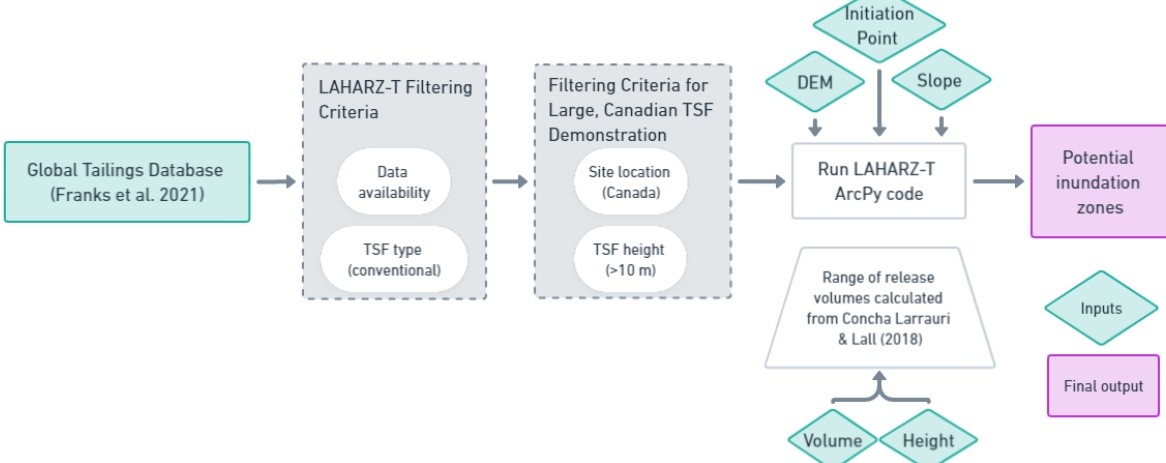

**Figure 2.** Methodology flowchart for the regional scale application of LAHARZ-T to a selection of large Canadian TSFs (dam height $\geq$ 10 m) as listed in the GTD from Franks et al. [38]. The methodology can be modified to fit user needs.

### 3.3.2. LAHARZ-T Input Generation

The LAHARZ program requires several inputs: a DEM, slope, cone apex, initiation point and release volumes. The 30-m ALOS PALSAR Global Radar Imagery [37] and the 30-m SRTM [35] were used as the DEM sources for this demonstration. Both satellite data are freely accessible and provide full and continuous coverage of Canada with a 30 m spatial resolution. For the purposes of high-level inundation mapping using LAHARZ-T, the DEM resolution should ideally not exceed 30 m and LAHARZ or LAHARZ-T cannot support DEMs with very fine resolutions of 2 m or less [40]. For site level hazard evaluation using LAHARZ-T, high-resolution DEMs of 5 m to 10 m may improve the model output [40]. For the purposes of portfolio or regional scale mapping, high-resolution DEMs are less relevant. High resolution DEMs decrease in utility further when evaluating the degree of interaction between inundation areas and regional or global datasets of potential downstream hazards, as these datasets are seldom finer grained than a 30 m DEM.

The cone apex, derived from Schilling [30] for TSFs, is the highest point of elevation on the facility. The slope is then calculated from the cone apex to the lowest point of elevation on the facility, perpendicular to the embankment. Predicting the initiation point of a TSF breach is challenging, as an initial breach location can be influenced by a number of site-specific factors, such as the geometry of the TSF and its foundation, the properties of impounded materials or the failure mechanism/mode [41]. Consistency in initiation point selection, independent of the TSF geometry, is critical for application efficiency and precision. Therefore, for high-level, regional applications, the lowest point of elevation on the dam toe can be selected as the initiation point. This selection prevents the program from erroneous outputs of the flow simulations running back into the tailings pond, as opposed to outward into the environment. However, the proposed methodology is no substitute for engineering judgement and this methodology should only be adopted as a high-level solution for large datasets. For small scale or local scale application, potential breach initiation points should be reviewed with detailed numerical modelling with the support of site-specific information. The LAHARZ-T program has the capability to run nine potential initiation points, which is useful for testing multiple failure scenarios efficiently. However, in the case of high-level regional aggregate consequence assessments, running multiple failure scenarios for a single facility may lead to double-counting errors of potential downstream impacts.

### 3.3.3. Release Volume

The final volume of tailings released from a facility is a critical input variable in TSF breach-runout assessments [42]. Historically, the final release volume of TSF failures has ranged widely from below 10% to over 90% of the total impounded volume depending on case-specific factors such as the TSF size and configuration, local and downstream topography, trigger variables, failure modes and tailings properties [4,13]. LAHARZ has the capability of running up to seven release volumes simultaneously. This functionality is leveraged to incorporate the uncertainty associated with the release volume of a TSF. Griswold et al. [31] advocated the use of a range of hypothetical flow volumes that span orders of magnitude for providing graphical outputs that depict uncertainty in the predicted inundation areas, in the form of nested hazard maps. In this study, Concha Larrauri & Lall's [23] equation (Table 1) is used to calculate the mean release volume (MP) of each facility. To account for the uncertainty in the volume estimates and show a range of scenarios within prediction intervals, an upper prediction (UP) and lower prediction (LP) are constrained to the upper and lower 50% confidence interval around the $V_F$ regression model, respectively. The UP and LP bounds in the Concha Larrauri & Lall [23] release volume equation offer a practical way to estimate a range of potential release volumes via a freely accessible calculation app, which outputs uncertainty-bounded release volumes for any given TSF storage volume. For future applications, the prediction interval can be increased to incorporate a wider range of possible outcomes. For example, in the most conservative approach, the user may input the total storage volume of the facility as an upper limit. Moreover, as updated models for release volume predictions are developed, such as Piciullo et al. [7] and Rana et al. [13], the model can accommodate these evolutions. Given the importance of incorporating uncertainty around the release volume, the final output of the LAHARZ-T model includes nested hazard maps based on the modelled release volumes (LP, MP, UP).

## 4. Results

### 4.1. Model Calibration

Figure 3 shows the log-log regression line for Zone 1 cross-sectional area as a function of total release volume. The regression with a specified 2/3 slope (consistent with the relationship presented earlier in Equation (2)) plots within the 95% confidence interval of the best-fit regression, supporting the hypothesis that this same scaling relationship is valid in the case of tailings breach data. The 95% prediction interval for the specified 2/3 slope regression is also plotted in Figure 3. The lower and upper 95% prediction intervals indicate the level of uncertainty associated with the prediction of inundation areas using this empirical approach. Table 4 shows the regression results for the maximum cross-sectional area vs. release volume for the best fit line and the fitted 2/3 slope regression model. The following equation was obtained in power-law form for the specified 2/3 slope regression model:

$$B = 0.1V^{2/3} \tag{3}$$

**Table 4.** Statistical results of the regression analysis for the best-fit and specified 2/3 slope models.

| Parameter | Best-Fit Regression | Specified 2/3 Slope |
|---|---|---|
| Slope ($\alpha$) | 0.53 | 0.67 |
| Intercept of line at log V = 0 (Log($\beta$)) | −0.14 | −0.89 |
| $\beta$ | 0.72 | 0.1 |
| Number of data, $n$ | 32 | 32 |
| Standard error of model, $\sigma$ | 0.39 | 0.40 |
| Standard error of volume coefficient | 0.08 | NA |
| Standard error of intercept | 0.46 | 0.07 |
| Coefficient of determination, $R^2$ | 0.58 | 0.56 |

[1] The power-law form of the equation: $A = (\beta) V^{\alpha}$; The linear form of the equation in log-log scale: Log($A$) = $\alpha$ Log($V$) + Log($\beta$). For $\alpha = 2/3$, $\beta = c_B$ coefficient in Equation (2).

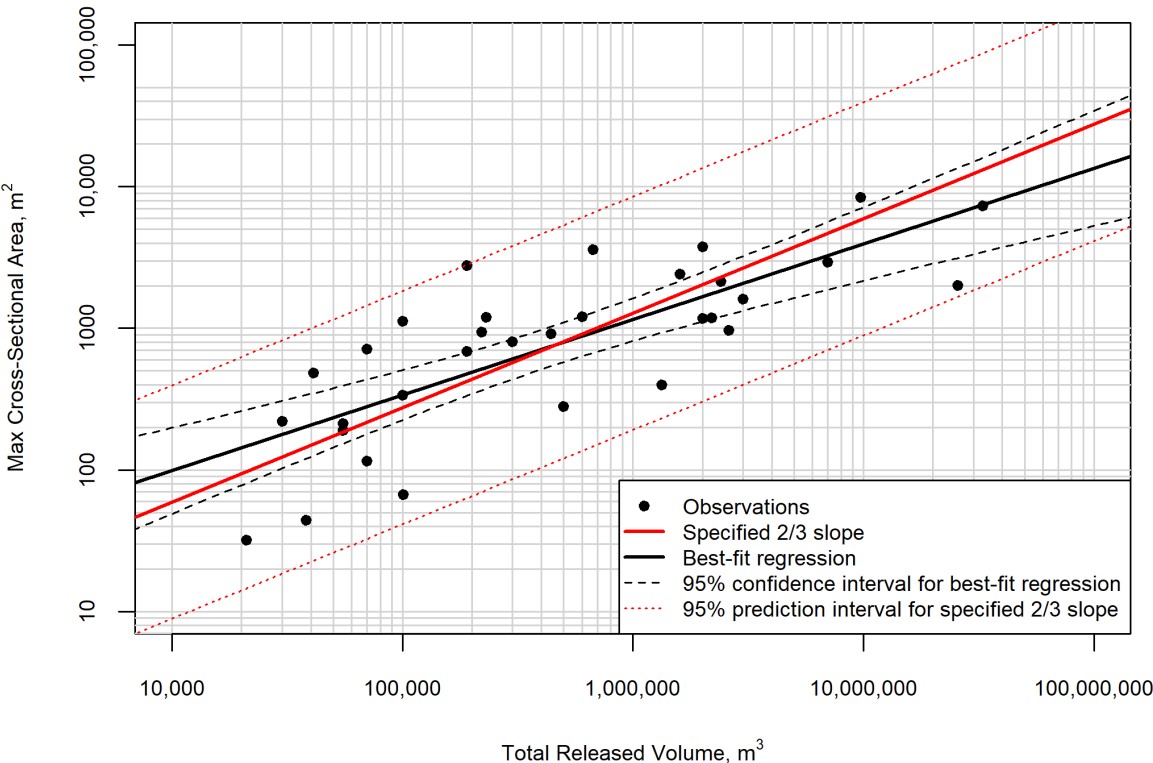

**Figure 3.** Log-log scatter plot of Zone 1 cross-sectional area vs. total release volume for 32 tailings flows. The specified 2/3 slope regression line (in red) is fitted to the data and the 95% confidence intervals associated with this trend are shown. The best-fit regression line (in black) and the 95% prediction intervals (red dashed lines) of the best-fit regression are plotted for comparison.

### 4.2. Model Validation

The modelled predictions based on the leave-one-out analysis are compared to the observed inundation areas and Zone 1 runout extents in Figure 4. The LAHARZ-T model prediction of Zone 1 runout distance is less robust than Zone 1 inundation area. In the comparison between modelled and actual runout distance and planimetric area, we observe a wider spread around the line of perfect fit in the case of runout. In our dataset, 84% of failures fall within one order of magnitude of the line of perfect fit for either the runout distance or Zone 1 inundation area. Two unconfined flow cases (P. 9 and P. 31) fall outside an order of magnitude prediction, skew the normalized indices heavily and are noted as outlier events (Table 5). Unconfined flows exhibit a wider spread of normalized indices compared to channelized flows.

**Table 5.** Cases where the predicted area or volume falls outside one order of magnitude from observed values.

| Case | Failure | Confinement | Exceedance |
|------|---------|-------------|------------|
| P. 2 | Cerro Negro | Unconfined | Area |
| P. 9 | Stancil | Unconfined | Area |
| P. 20 | Ajka | Channelized | Runout distance |
| P. 28 | Mishor Rotem | Channelized | Area/Runout distance |
| P. 31 | Cadia | Unconfined | Runout distance |

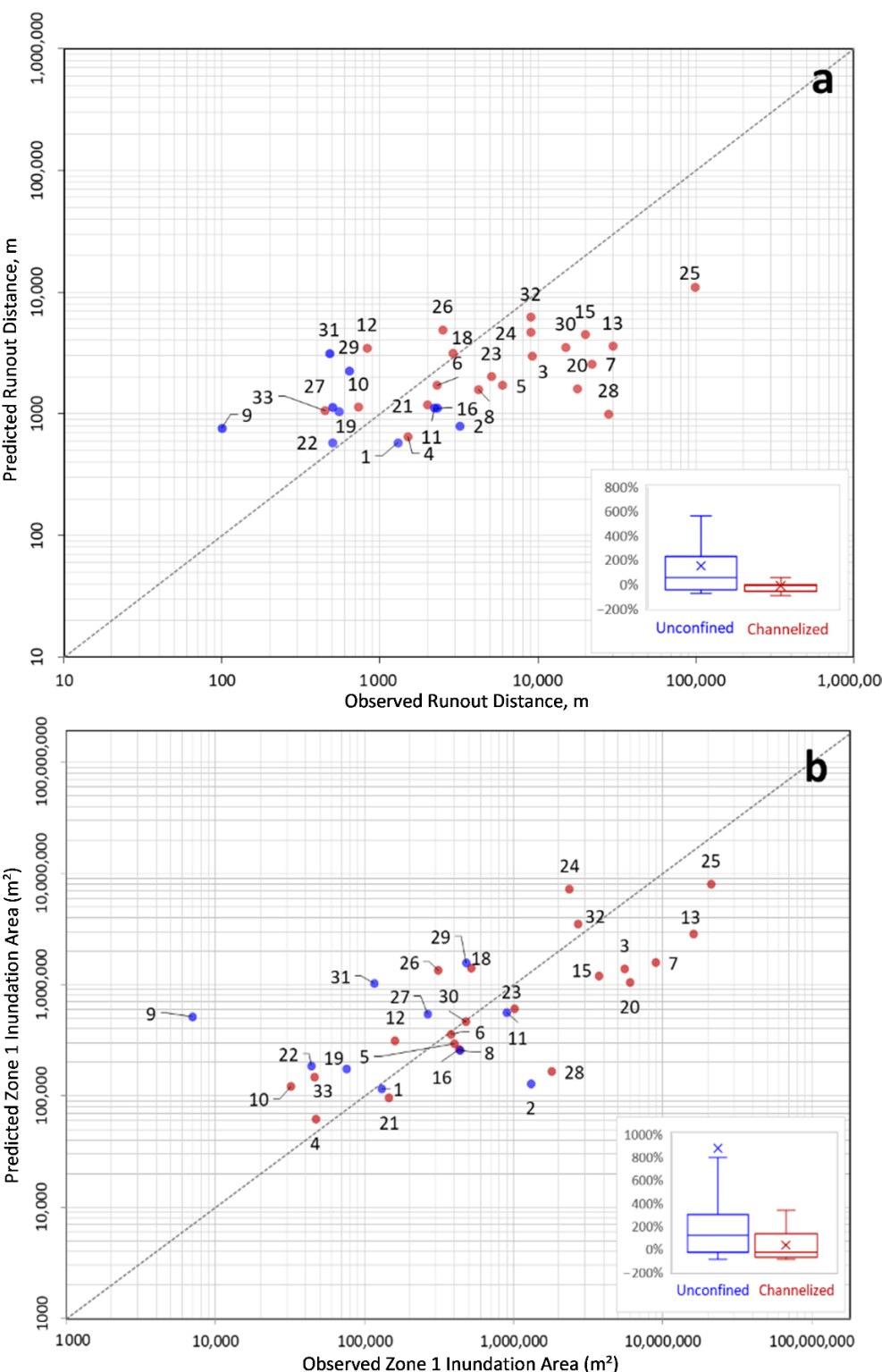

**Figure 4.** (**a**,**b**). Log-log scatter plot of LAHARZ-T predicted vs. observed Zone 1 runout distance (**a**) and predicted vs. observed Zone 1 inundation area (**b**). The insets show the spread of normalized indices for channelized flows (red) and unconfined flows (blue). The line of perfect fit is plotted (grey dashed line) for comparison. Stancil (P. 9, >7000%) was removed from the normalized indices area plot for legibility. See Table 3 for event numbers.

Table 5 lists the cases where the predicted runout distance or inundation area falls outside of one order of magnitude from the observed values. The Fundão (P. 25) and Stancil

(P. 9) cases, which represent the maximum and minimum observed distance/area extents in the database, respectively, are the most influential observations in Figure 4a,b. The Mishor Rotem (P. 28) case is strongly under-predicted in Figure 4b. As noted by Ghahramani et al. [4], the extreme observed runout distance of this case could be attributed to (i) high free water content, (ii) the narrow, dry channel path in a desert environment with no physical obstacles to flow, and (iii) a potential increase in flow volume by entrainment. LAHARZ-T tends to underpredict the runout distance and inundation area of a high-volume failure events and overpredict the runout distance and inundation area of low volume events. With the exception of the cases listed in Table 5, the vast majority of the cases fall within one order of magnitude of observed values, which suggests that the model is acceptable for use in regional scale modelling or high-level screening tests.

*4.3. Demonstration of the Semi-Automated LAHARZ-T Model for Regional Use: Canadian Application*

Although the LAHARZ-T model was calibrated and validated on individual failure events, the primary application of the model in the present research is not to predict potential tailings flow impact zones at the local scale. The selected approach rather justifies and favours a broader, regional scale application, where indications of potential hazards to elements at risk can be derived in the sense of a first-order assessment. As outlined in Section 3.3, 46 large Canadian TSFs from the GTD were selected for a regional scale demonstration of the LAHARZ-T model. The subset of Canadian sites are well-distributed across Canada, with notable clusters in north-central British Columbia and the Ontario-Quebec border.

Figure 5 graphically illustrates the modelled inundation area corresponding to the mean estimated release volume at each site and the facility's GTD-listed, "self-reported" (i.e., reported by the respective mining company within the GTD questionnaire) consequence classification of each TSF that was modelled. The authors note that in the original GTD questionnaire and the Franks et al. [38] database, companies were requested to disclose a facility's hazard categorization. For clarity and consistency with current standards, the authors use the terminology "consequence classification", which aligns with the Canadian Dam Association (CDA) ratings for tailings dams [43] and the terminology used in Franks et al. [38].

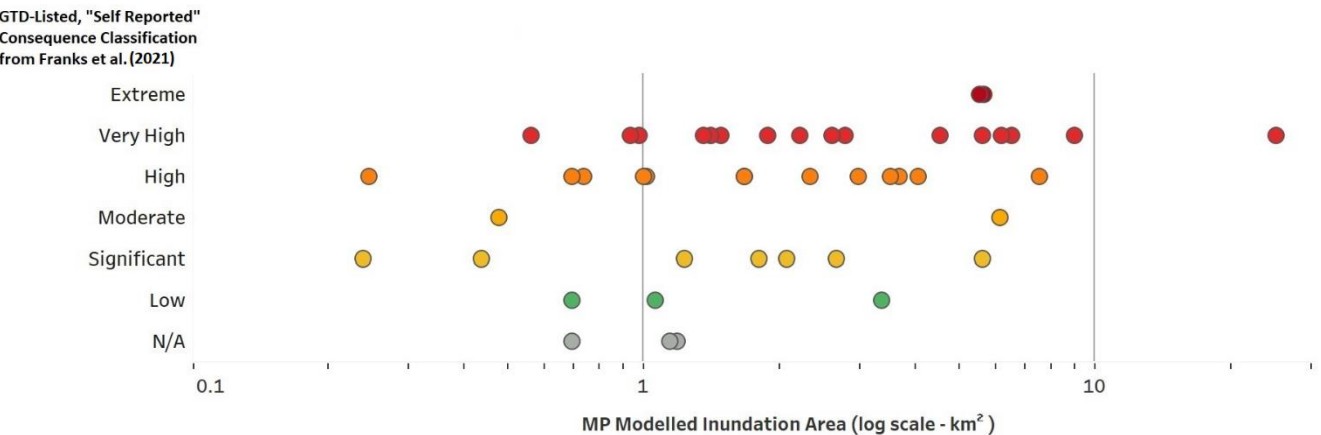

**Figure 5.** Log-scatter plot of the MP modelled Zone 1 inundation areas in km$^2$ for the 46 Canadian TSFs modelled with LAHARZ-T and the TSFs' corresponding GTD-listed consequence classifications. The modelled inundation areas reflect the mean release volume (MP) calculated for each facility. The consequence classifications of each facility are "self-reported" and sourced from the GTD [38].

The average modelled inundation areas of the 46 TSFs were 2.2 km$^2$ for the LP volume, 3.1 km$^2$ for the MP volume and 4.8 km$^2$ for the UP volume. Most of the modelled TSFs in British Columbia have high to extreme GTD-listed consequence classifications and modelled inundation areas above the Canadian average, and most of the modelled TSFs in Ontario and Quebec have high or lower consequence classifications and modelled

inundation areas below the Canadian average. As shown in Figure 5, there is a loose visual correlation between the modelled inundation area estimates and the GTD-listed consequence classes across the whole dataset. A strong correlation would not be expected, as a large modelled inundation area does not necessarily reflect a large potential consequence downstream (e.g., if there are no population centers, environmentally sensitive habitats or infrastructure in the area).

As discussed in Section 2, the LAHARZ-T model is calibrated for Zone 1 impacts only; therefore, the interaction of potential inundation areas with waterbodies (i.e., Zone 2 impacts) affects the validity of the inundation area and runout distance output from the model (see Section 5.2—Limitations). To evaluate the number of Canadian sites with interactions with waterbodies, the inundation area polygons were geoprocessed against geospatial data of Canadian rivers and lakes [44]. Of the 46 Canadian sites, 21 inundation polygons (46%) have some degree of interaction with a waterbody.

A promising direction for future research is to overlay the modelled outlines of downstream inundation areas with potential elements at risk such as protected areas, conservation areas or critical infrastructure, as illustrated by Innis et al. [33]. These overlays can then be used to better inform mining stakeholders of regional, national or portfolio-wide consequence profiles.

## 5. Discussion

### 5.1. Implications

This study is an extension of the recent work to advance release volume-based empirical relationships for tailings flows [4,13,24] and provides a semi-automated regional hazard mapping tool for TSF breach assessment (LAHARZ-T). The results from Table 4 indicate that Equation (2) is statistically justifiable for the relationship between total release volume and cross-sectional inundation area of tailings flows, given that the specified 2/3 slope line falls within the 95% confidence interval curves for the best-fit regression, and represents a useful scaling relationship for practical applications. The validation work showed that the LAHARZ-T model can predict Zone 1 runout distance and Zone 1 planimetric inundation area of tailings flows within one order of magnitude of observed values in 84% of the historical cases in the dataset. LAHARZ-T is therefore a useful tool for hazard mapping projects at the regional, national or global scale. Semi-empirical equation derived tools, such as this one, may also be used for low-cost screening level assessments at mine sites for single dams, multi-dam TSFs and portfolio prioritization. However, as stated in Section 1 (Introduction), LAHARZ-T should not be used deterministically. The proposed methodology provided in Section 3.3.1 for the regional application of LAHARZ-T allows for the inclusion of uncertainty around the release volume of potential failures. At least one level of uncertainty should be included when using LAHARZ-T for forward-looking analyses. Uncertainty around the mobility coefficients ($c_a$ and $c_b$) may also be written into the underlying LAHARZ-T code to further improve the stochastic functionalities of the model.

The introduction of LAHARZ-T into the field of TSF breach-runout assessment is timely and significant due to the sparsity of semi-empirical tailings flow models and the lack of models capable of undertaking regional scale hazard mapping projects. The work by Owen et al. [45] examining the local risk of TSFs to environmental and social variables on a global scale presented one of the first analyses of TSF risks at a large scale; however, the analysis used simple buffering techniques with limited information on the TSF types. The LAHARZ-T model offers an improvement to the Owen et al. [45] buffering method, while working towards aligned goals on increasing transparency between mining industry stakeholders and improving the understanding of aggregate ecosystem and social pressures from mining.

The expansion of LAHARZ-T to national-scale modelling, which is contingent on continual improvements in disclosures, may improve the contextual understanding of the risk from potential TSF failures to Canadian environments. For example, the degree

of interaction between potential tailings flows and waterbodies would be of particular concern in Canada.

### 5.1.1. Implications for Portfolio-Level Modelling for Institutional Mining Investors

Institutional mining investors influence tailings management and environmental, social and governance (ESG) practices. Since the 2019 Feijão failure, institutional investors have taken lead roles in establishing practices and initiatives to improve understanding and transparency of social and financial risks associated with TSFs, such as the GTD [46]. Despite these initiatives, there remain few tools for tailings-related performance of mining companies for use by investors. Previous research on the role of institutional mining investors in improving tailings management showed that the current GTD may not support investor action to improve TSF management to the degree that it was initially intended [17]. Continued investor action for improvement in tailings and ESG criteria [47,48] suggests that there is interest and motivation to act as engaged shareholders on these issues.

Portfolio-level tools using a simple, high-level model, such as LAHARZ-T, may be useful for institutional mining investors to transform data from initiatives such as the GTD into actionable information. Future work may include transforming the LAHARZ-T model into a portfolio-level, tailings-specific ESG ranking tool for use by institutional investors, investor stakeholders and ESG investment ranking services, such as Sustainalytics, or industry-wide organizations, such as ICMM or the Mining Association of Canada. Similar ranking tools exist for political risks across investment portfolios [49], water scarcity or flooding risk [50,51] and exposure to climatic extremes [18]. Institutional mining investors use ranking tools in a range of ways, from fundamental analysis and company valuation to active ownership assessment, such as company engagement and voting [52].

### 5.1.2. Implications of Regional-Level Scale Modelling for Policymakers

The risks of mining infrastructure failures have been highlighted by many governing bodies of countries with recent TSF failures. For example, the Mount Polley TSF failure in British Columbia, Canada, led to the tightening of legal requirements around best practice guidelines. However, despite the improvements in regulation, compliance verification and enforcement has proved challenging for several reasons including a lack of data, data discrepancy in reporting and insufficient resources at the ministry level relative to the abundance of TSF sites [20,21]. As noted in the most recent audit from 2021, no procedure exists to address prioritization of facilities and provide a reflection of perceived risks associated with different types of mines and downstream hazards [20].

A model such as LAHARZ-T could be applied at the provincial or national level to assist in mine site audits, with the ability to quickly identify areas of higher social, financial or environmental hazard. Recent work in the industry has begun to formulate these frameworks around the prioritization of downstream hazards and geotechnical factors [53].

### 5.2. Limitations of LAHARZ-T

There are inherent limitations associated with simple, high-level semi-empirical models (such as LAHARZ-T) compared to detailed numerical models. Based on the present study, these limitations and the implications of these limitations are summarized as follows:

- The output of LAHARZ-T does not include flow evolution parameters such as flow velocity or accumulation thickness. In cases where flow depth and velocity are necessary for decision making such as in emergency planning, numerical models should be used.
- Field conditions related to the rheological properties of the tailings were not directly considered in the calibration of the mobility coefficients. Highly saturated tailings flows may not be adequately modelled using LAHARZ-T.
- The planimetric and cross-sectional mobility coefficients ($c_A$ and $c_B$) represent a global sample of failures and are not further classified into coefficients based on downstream confinement (Table 1, Rana et al. [13]) This allows for the model to have a broader, multi-terrain application supporting regional or portfolio approaches. However, this

may result in the model responding less accurately where flows are unconfined. As discussed in Rana et al. [13] and Ghahramani et al. [4], additional information on historical tailings flows are required to improve the relationship between inundation area and downstream confinement. As more data becomes available, the mobility coefficients can be further refined.

Figure 6 demonstrates different tailings flows modelled using LAHARZ-T. Several factors influence the output, including the transition between Zone 1 and Zone 2 flow, confinement of the downstream terrain and the quality of the DEM. The interaction of modelled inundation areas with water bodies results in "spikey," irregular outputs. This is due, in part, to the geometric distortions related to the interaction between water bodies and radar or laser-based data collection [54,55]. However, as discussed in Section 2, the underlying empirical equations are calibrated to predict only Zone 1 impacts, and the transition of land to water also corresponds to a transition between Zone 1 and Zone 2 impacts; therefore, the flows within water bodies (i.e., Zone 2 impacts) are uncalibrated. Despite this limitation, the interaction itself between the potential tailings flow inundation area and the water body is a valuable insight and can contribute to a better understanding of the risk to water bodies from TSFs, as demonstrated in Section 4.3.

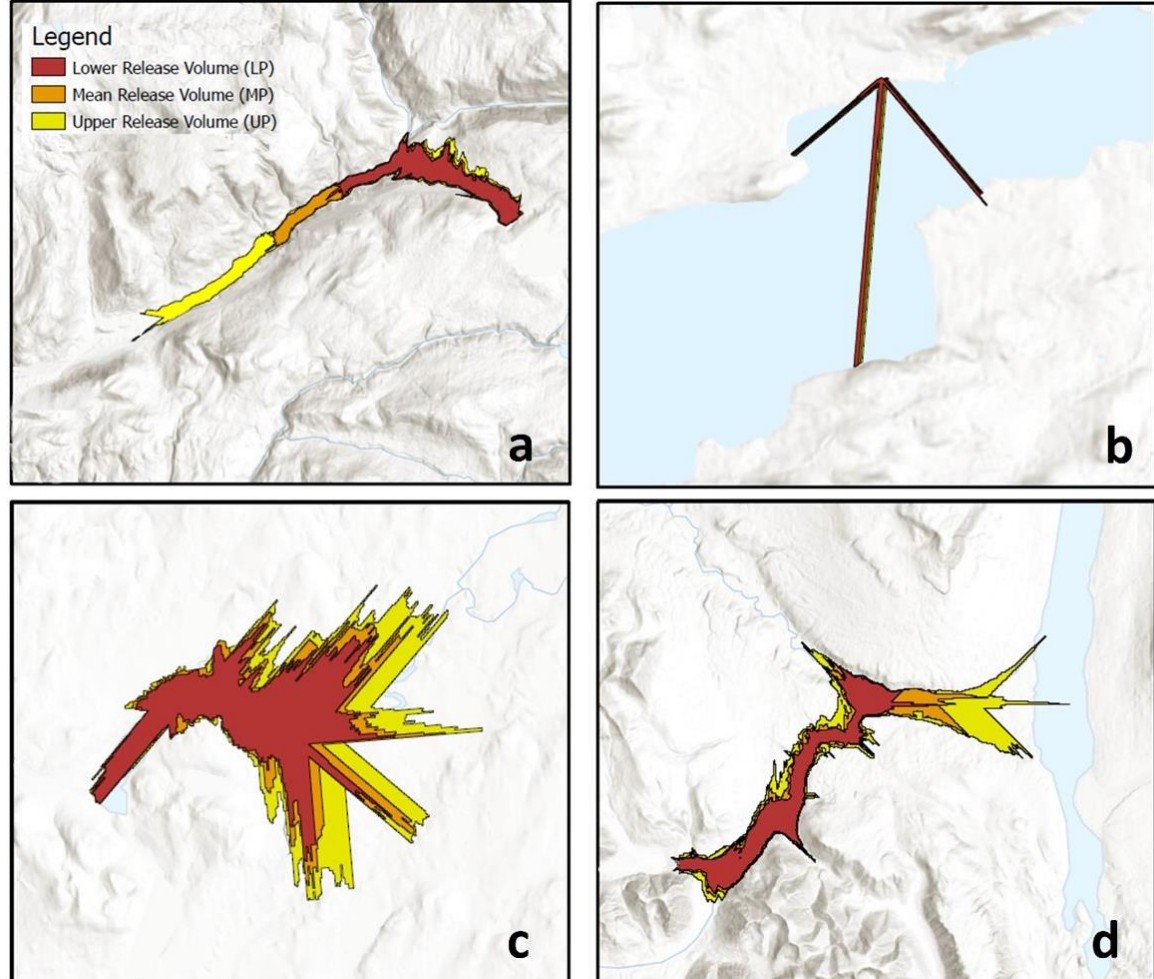

**Figure 6.** Select results of the Canadian LAHARZ-T demonstration, including a channelized flow (**a**), an interaction with a water body (**b**), an unconfined flow (**c**), and a channelized flow with a water body interaction (**d**).

The effects of downstream topographic confinement (or lack thereof) on modelled results is discussed in research on the application of LAHARZ to natural flows [56,57]. A

lack of well-defined channel banks and unconfined downstream terrain makes it challenging to accurately delineate cross-sectional areas. The differences between a channelized flow output and an unconfined flow are also demonstrated in Figure 6, as well as the different levels of lateral edge spikiness associated with several of the LAHARZ-T outputs. The spikiness worsens in outflows with waterbody-related DEM inconsistencies and areas with undefined channel banks. In previous applications of LAHARZ to debris flows and lahars, approaches have been developed to limit the spikiness of lateral edges of the outputs [40,56,58]. However, due to the regional scale application and the simultaneous modelling of dozens of facilities, these approaches are considered to be less practical in the context of the present study. Given the limitations of LAHARZ-T, caution must be used when modelling TSFs near waterbodies and for regional work where unconfined terrain is present.

## 6. Conclusions

Semi-empirical modelling of inundation areas, having the advantage of low cost, user accessibility and high efficiency, can play a key role in TSF risk management. The central goal of this study was to adapt the LAHARZ model, established originally for lahars, to tailings flows, to provide researchers and practitioners with a relatively simple risk mapping tool (LAHARZ-T). In this study, data collected from 32 historical tailings flows between 1965 and 2019 were spatially analyzed to develop power-law equations to relate the cross-sectional area of a tailings flow to its release volume, and to modify the LAHARZ program for applicability to TSF sites.

The relationship between cross-sectional area and release volume was found to be $B = 0.1V^{2/3}$. The results support the hypothesis that tailings flows follow the same general relationships between flow volume and inundation area as those for natural flow-like landslides, including lahars, debris flows and rock avalanches. The relationship was validated by a cross-validation exercise and outlier case identification. For most failures and all non-outlier cases, LAHARZ-T predicted the Zone 1 runout distance and planimetric inundation area of TSF failures within one order of magnitude. Thus, the primary application of LAHARZ-T favours broader, regional scale application. To demonstrate the methodology, usefulness and limitations of LAHARZ-T on a regional scale, a sample of 46 large Canadian TSFs were selected for modelling.

This research produced two code outputs that are available via request to the lead author: (1) the LAHARZ-T semi-automated ArcPy code; and (2) a batch raster to polygon automated ArcPy code. Further work is underway to transform the LAHARZ-T model and related codes into a tool for use by institutional mining investors and government mine auditors.

Although the variability in tailings properties and site characteristics cannot be perfectly incorporated or modelled, the LAHARZ-T model offers promise for high-level hazard assessment, addressing the increasing call from many mining stakeholders for more robust and multi-scale understanding of the risks from potential TSF failures.

**Author Contributions:** Conceptualization, S.I., N.C.K. and S.M.; methodology, S.I., N.G. and N.R.; software, S.I.; validation, S.I. and N.R.; regional analysis, S.I.; data curation, N.R., N.G. and S.I.; writing—original draft preparation, S.I., N.G. and N.R.; writing—review and editing, S.I., N.R., N.G., N.C.K., S.M., W.A.T. and S.G.E.; visualization, S.I.; supervision, N.C.K., S.M., S.G.E. and W.A.T. All authors have read and agreed to the published version of the manuscript.

**Funding:** This work was partially funded by the British Columbia Graduate Scholarship and a PGS-D from the Natural Sciences and Engineering Research Council of Canada (NSERC) issued to S.I. [PGSD3-#559876], fellowships from the University of British Columbia (N.G. and S.I.) and University of Waterloo (N.R.), as well as a Collaborative Research and Development grant [CRDPJ 533226-18] from NSERC, in partnership with Imperial Oil Resources Inc., Suncor Energy Inc., BGC Engineering Inc., Golder Associates Ltd., and Klohn Crippen Berger. N.C.K. also acknowledges that her research program is supported by an NSERC Tier 2 Canada Research Chair.

**Data Availability Statement:** The LAHARZ-T ArcPy and additional ArcPy regional mapping code are available by request to the author(s).

**Acknowledgments:** The authors would like to acknowledge Jared Lynd for data cleaning efforts, Andrew Mitchell for input on the model validation approach and Salman Haider for support on improving the ArcPy codes.

**Conflicts of Interest:** The authors declare no conflict of interest.

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
