# Peer review of "The Development and Demonstration of a Semi-Automated Regional Hazard Mapping Tool for Tailings Storage Facility Failures"

_resources, doi:10.3390/resources11100082_

Round 1
Reviewer 2 Report
Relationship caused by the author in view of the tailings dam downstream flood area this crucial question, has carried out large-scale regional scale of TSF risk assessment semi-automatic drawing program research, based on the development of surveying and mapping geographic information data of collecting the GIS semi-automatic hazardous area partition and risk assessment procedures and code, this paper proposes a new risk assessment tools LAHARZ-T, Filtered after inundation risk classification of the tailings dam, through modeling to predict the tailings after dam flow potential flood flooded area, stroke distance and path to reduce the risk after the dam downstream, such as in Canada, 46 TSF and good application effect, to reduce the potential dangers from downstream of the tailings dam area risk recognition and has a certain guiding significance for the prevention of However, throughout the article, there are the following problems:
1. On the whole, the expression of the article is verbose, not concise and clear enough, and not rigorous enough in logic.
2. The proposed new model is an empirical model based on Equations 1 and 2, which ignores the accumulation thickness caused by tailings flow, the evolution of flow velocity, the variability of downstream geomorphic conditions and rheological properties, etc., so the prediction accuracy is slightly insufficient. It is suggested to supplement the results of follow-up examination combined with field conditions to reduce the influence of uncertain factors and improve persuasiveness.
3. The serial number of the second 2.1 should be 2.2.
4. Model verification is only targeted at a specific region by using the method of statistical probability (the power law function relationship derived from this), which is not universal, and the factors not considered are not clearly explained in the model. Do we need to introduce new variables for comprehensive consideration? Do you need to optimize existing parameters? Whether it is suitable for other regions is unknown.
5. In terms of parameter values, the author directly cites the research results of Ghahramani et al., and determines that CA is 80, and only the cross-sectional flow coefficient CB is calibrated and optimized. Is it reasonable?
6. When using LaharZ-T modeling, the prediction accuracy is relatively high compared with that of zone 1. However, when considering TSF analysis near water bodies, especially the transition from land to water, how to consider the flow and transition between water bodies? How to calibrate? To what extent is it affected by unrestricted geomorphological conditions downstream? How to quantify it? These will more or less affect the accuracy of its prediction.
7. Lack of sensitivity analysis and discussion of influencing factors, and suggest to supplement.
Round 2
Reviewer 1 Report
All the comments have been well addressed and the paper was well revised.
Reviewer 2 Report
You revised your paper very well.